# HypoxaMIRs: Key Regulators of Hallmarks of Colorectal Cancer

**DOI:** 10.3390/cells11121895

**Published:** 2022-06-11

**Authors:** Jossimar Coronel-Hernández, Izamary Delgado-Waldo, David Cantú de León, César López-Camarillo, Nadia Jacobo-Herrera, Rosalío Ramos-Payán, Carlos Pérez-Plasencia

**Affiliations:** 1Genomics Laboratory, The National Cancer Institute of México, Tlalpan, Mexico City 14080, Mexico; izz.waldo11@gmail.com (I.D.-W.); dfcantu@gmail.com (D.C.d.L.); 2Functional Genomics Laboratory, Biomedicine Unit, FES-IZTACALA, UNAM, Tlalnepantla 54090, Mexico; 3Posgrado en Ciencias Genómicas, Universidad Autónoma de la Ciudad de México, Mexico City 03100, Mexico; genomicas@yahoo.com.mx; 4Biochemistry Unit, Institute of Medical Sciences and Nutrition, Salvador Zubirán, Tlalpan, Mexico City 14080, Mexico; nadia.jacoboh@incmnsz.mx; 5Facultad de Ciencias Químico Biológicas, Universidad Autónoma de Sinaloa, Culiacan City 80030, Mexico; rosaliorp@uas.edu.mx

**Keywords:** hypoxaMIR, angiogenesis, HIF-1α, microRNAs, metastasis, transcription factor, miRNA network, miRNA regulation

## Abstract

Hypoxia in cancer is a thoroughly studied phenomenon, and the logical cause of the reduction in oxygen tension is tumor growth itself. While sustained hypoxia leads to death by necrosis in cells, there is an exquisitely regulated mechanism that rescues hypoxic cells from their fatal fate. The accumulation in the cytoplasm of the transcription factor HIF-1α, which, under normoxic conditions, is marked for degradation by a group of oxygen-sensing proteins known as prolyl hydroxylases (PHDs) in association with the von Hippel-Lindau anti-oncogene (VHL) is critical for the cell, as it regulates different mechanisms through the genes it induces. A group of microRNAs whose expression is regulated by HIF, collectively called hypoxaMIRs, have been recognized. In this review, we deal with the hypoxaMIRs that have been shown to be expressed in colorectal cancer. Subsequently, using data mining, we analyze a panel of hypoxaMIRs expressed in both normal and tumor tissues obtained from TCGA. Finally, we assess the impact of these hypoxaMIRs on cancer hallmarks through their target genes.

## 1. Introduction

Colorectal cancer (CRC) is the third most common cancer, and represents the fourth leading cause of cancer mortality worldwide. In 2018, more than 1.8 million cases were diagnosed, and 880 thousand death cases were reported worldwide [1]. The most common risk factors associated with CRC development include genetic predisposition, diet, alcohol and tobacco consumption, age, and inflammatory bowel disease. Molecular mechanisms related to CRC progression are microsatellite and chromosomal instability, CpG island methylator phenotype, oncogene and tumor suppressor gene mutations [2]. Furthermore, variations in microRNA expression profiles [3] promote the formation of aberrant crypt foci and polyps [4], changes in the tissue microenvironment [5], the establishment of carcinoma [6], angiogenesis [7], and, finally, metastasis [8,9]. 

Although the multistep progression and the main mutated oncogenes are widely known, the molecular events that underly each step of CRC progression are not yet completely understood due to the high rates of genomic instability and intra-tumoral heterogeneity that characterize this type of cancer [10]. Therefore, several groups are actively searching for gene expression profiles in CRC from different perspectives; for instance, response to therapy [11], pathway analysis [12], and miRNA profiles [13]. Among these, we encounter hypoxia-regulated genes, given the importance of hypoxia in the induction of proliferation, cell invasion, and angiogenesis [14,15].

Hypoxia itself is an extensively studied phenomenon [16] whose response is coordinated by a family of transcription factors, collectively known as hypoxia-inducible factors (HIFs) [17]. These factors are overexpressed in multiple types of tumors, and both local and distant metastases [18]; consequently, genes under transcriptional control are overexpressed [19]. HIFs are relevant in maintaining malignancy, particularly in CRC [20]. However, the prognostic value of HIF expression by itself does not seem to be as relevant as the gene network regulated by HIF activity. For instance, HIF overexpression and high VEGF protein detection are associated with the vascular invasion of colorectal carcinoma and tongue squamous cell carcinoma [21], serving as prognosis biomarkers [22], while others disregard its importance as a biomarker [23,24]. However, some HIF-α hypoxia-regulated genes, such as VEGF-A, Smad7, Jun, IL-8, CXCR-4, PDGF-A, TGF-A, or ANGPTL-4, are considered markers of CRC metastasis and poor prognosis [14]. Therefore, even though the relevant role of HIF-α activation in cancer development is not a *bona fide* molecular marker, genes that are under its transcriptional control, such as those formerly mentioned, may be valid as biomarkers or new therapeutic targets. Therefore, in this work, we focus on describing the role of HIF-1α in regulating through the transcription of microRNAs called hypoxaMIRs, each representing the hallmarks of CRC, providing a broader picture of how these types of non-coding RNAs are involved in the multistep progression of this type of cancer.

## 2. Hypoxia-Inducible Transcription Factors

HIFs are a family of heterodimeric transcription factors conformed by one of three possible isoforms of an O2-labile α subunit (HIF-1α, 2α, and 3α), and a second stable HIF-1β subunit. HIF-1α is ubiquitously expressed in all human cells; HIF-1α subunits dimerize with HIF-1β through their HLH (helix–loop–helix) and PAS (Per–Arnt–Sim) domains to play a role as a transcription factor. The HIF-1α subunit contains an N-TAD (N-terminal transactivation domain) that overlaps with the ODD domain (oxygen-dependent degradation) associated with protein stability, while C-TAD interacts with the transactivator protein CBP/P300. In normoxic environments, conserved proline residues of HIF-1α (Pro 402 and Pro 564 localized in ODD domain) are hydroxylated by prolyl-4-hydroxylases (PHDs) through molecular oxygen and 2-oxoglutarate as substrates for their activity [25]. Once HIF-1α is hydroxylated, it is recognized and marked for proteasome degradation by Von Hippel-Lindau (pVHL) tumor suppressor protein, an E3 ubiquitin ligase. Conversely, under hypoxic conditions, PHD activity is diminished, so they cannot hydroxylate HIF-1α and, as a consequence, HIF-1α is stabilized and translocated into the nucleus, where it heterodimerizes with the HIF-1β subunit. HIF-1α recognizes and binds to the consensus sequence G/ACGTG, better known as HRE (hypoxia response elements) [26], which is present in the promoter region of genes that promote the maintenance of cancer hallmarks, such as the epithelial–mesenchymal transition (EMT), angiogenesis, metastasis, and drug resistance (Figure 1). 

EMT is one of the central mechanisms for the induction of invasion and metastasis [27]. In the EMT process, polarized epithelial cells lose their polarity and acquire migrating and invasive properties. HIF-1α mediates EMT by down-regulating epithelial markers (E-cadherin), and increasing the expression of mesenchymal markers (such as N-cadherin and Vimentin) [28]. HIF-1α transcribes ZEB1 (zinc finger E-box binding homeobox 1), which can bind to the E-cadherin promoter, and inhibits its transcription in CRC cell lines, suggesting a relevant role of HIF-1α in CRC [29].

Moreover, regulating the expression of many genes involved in tumor maintenance, it has been described that HIF-1α is able to promote drug resistance through metabolic reprogramming by increasing the transactivation of PDK1, LDHA, and BNIP3-L to switch from oxidative to glycolytic metabolism, an event known as the Warburg effect [17]. It has been reported that this metabolic rewiring prevents mitochondrial production of reactive oxygen species, thereby reducing the DNA damage and attenuating the antitumor effect of TKIs (tyrosin–kinase inhbitors) in several types of cancer [30,31,32,33]. Furthermore, in CRC cells treated with 5-FU (5-fluoruracil), the damage of mitochondria promotes the loss of their main source of energy, and, consequently, cell death is induced. In this scenario, HIF-1α, through the activation of the Wnt/β-catenin and PI3K pathways, stimulates the transcription of glycolytic genes, such as GLUT1, HK2, PKM2, and LDHA, restoring the lost energy, and promoting drug resistance to 5-FU [34].

CRC cells develop drug resistance by different mechanisms; in patients bearing the G12V mutation in the KRAS gene, it has been shown that high levels of HIF-1α are associated with resistance to Cetuximab. When a HIF inhibitor is used (PX-478), the cells were drug-sensitized, abating proliferation [35]. In addition, CRC-derived cell lines treated with DNMTs inhibitors indirectly diminished HIF-1α activity, leading to loss of resistance to oxaliplatin treatment [36]. Therefore, due to the large number of mechanisms involved in cancer establishment regulated by HIF-1α, it is important to describe their relevance in colorectal cancer.

## 3. HIF-1α and Its Relevance in CRC Development

HIF-1α is highly overexpressed in CRC at mRNA and protein levels; it is detected in adenomas and adenocarcinomas, and is frequently correlated with VEGF (vascular endothelial growth factor) overexpression, tumor vascularization, lymphatic invasion, disease stage, and overall survival. Immunostaining assays by several groups have shown that HIF-1α is present in 70–77% of CRC tumor cells, perinecrotic tissue, and blood vessels [37,38]. Current studies have shown the multiple roles that HIF-1α plays in CRC development and maintenance. HIF-1α promotes JMJD2B oncoprotein transcription in colorectal cancer-derived tumor cells, triggering a significant increase in cell proliferation and invasion through the overexpression of SLC2A1, SLC2A3, ELF3, UCA1, MET, NOV by removing trimethylation of H3K9 on their promoters [39]. Conversely, the inhibition of HIF-1α with sh-RNAs and lncRNAs avoids vasculogenic mimicry (a process in which cancer cells mimic endothelial cells by forming blood vessels), reducing their metastatic capabilities, and restoring E-cadherin expression in Lovo and HCT116 cells [40,41]. HIF-1α also promotes survival and chemoresistance in CRC through the direct modulation of survivin [42] and EZH2 [43]. In addition, mTOR/PP2A promotes cell survival via PHD2 inhibition, resulting in HIF-1α stabilization [44]. 

Wnt/β-catenin, another commonly overactive pathway in CRC that stimulates proliferation and migration, could be indirectly regulated by HIF-1α. In this pathway, the APC protein induces β-catenin degradation via the proteasome. Interestingly, the APC promoter contains an HRE element; thus, hypoxia levels allow HIF-1α to bind to this site, causing direct transcriptional repression in APC mRNA [45]. Moreover, HIF-1α overexpression induces β-catenin nuclear location, leading to high rates of proliferation, tumor growth, and radioresistance [46]. The mechanism by which HIF-1α promotes transcriptional repression is currently unknown; nevertheless, it has been reported that HIF-1α overexpression promotes transcriptional repression of several DNA repair genes, such as MSH2, MSH6, and NBS1, by promoter occupancy [47]. 

These examples demonstrate the relevance of HIF-1α in the hallmarks of colorectal cancer. However, the scope of this regulation is amplified due to the capability of HIF-1α activation of microRNAs transcription as direct targets, a process which will be discussed in depth in the following sections. 

## 4. HypoxaMIRs, miRNAs Regulated by HIF-1α

MicroRNAs (miRNAs) are non-coding short RNAs that target the 3′ untranslated region (3′UTR) of mRNAs, and cause their silencing by recruiting proteins that induce translational repression, mRNA deadenylation, and mRNA decay [48]. Approximately 1% of the transcriptome in eukaryotic organisms consists of miRNAs; thus, they play a key role as post-transcriptional regulators of gene expression [49]. miRNAs could regulate tumor suppressor genes, allowing the development of cancer; as such, they have been designated as oncomirs; their counterparts, miRNAs that regulate oncogenes negatively, are called miRNA tumor suppressors, or anti-oncomirs [50]. Several reviews concerning biogenesis, maturation, and their role in multistep carcinogenesis have been published.

Furthermore, HIF-1α plays a major role in the miRNAs’ biogenesis and maturation [51,52,53]. Previous reports have demonstrated that hypoxia activation decreases the levels of several proteins involved in microRNAs biogenesis, such as DGCR8, Exportin 5, DICER, TRBP, AGO1, and AGO2 [52], as well as modulating microRNA expression of mir-200 family [54] and mir-630 [55].

One of the main targets of HIF-1α is DICER, a key protein in microRNA processing. In colon tumors and HCT116 cells, HIF-1α recruits the ubiquitin ligase E3 parkin, forming a heterodimer that binds and ubiquitinates DICER, inducing its degradation by autophagosomes, thereby decreasing the expression of mir-200b [56]. In addition, in breast cancer cell lines under hypoxic conditions, DICER promoter can be hypermethylated by KDMA6A/B and EZH2, resulting in a mir-200b decrease, as well as EMT activation [57]. These findings are worth exploring to establish a putative role of HIF-1α participation.

Hypoxic conditions not only regulate DICER, but also play a crucial role in the maintenance of Argonaut proteins. HIF-1α promotes the expression of EIF2C4, which positively regulates Argonaut 4 (AGO4), inducing mir-18a, mir 107, mir-155, and mir-210 overexpression and angiogenesis induction [58]. Moreover, HIF-1α transcribes collagen type I prolyl-4-hydroxylase (cP4HI), which enhances the prolylhydroxylation and endonuclease activity of AGO2. Once AGO2 is hydroxylated, the processing of numerous microRNAs, such as mir-210, mir-21, mir-24, mir-221, mir-222, mir-25, mir-100, mir-15b, mir-23a, among others, is increased [59]. These data support the role of HIF-1α as a key regulator of miRNAs biogenesis in cancer (Figure 2).

HIF-1α plays a dual role in the regulation of miRNAs. HIF-1α directly regulates their biogenesis, and promotes the transcriptional activity of miRNAs with HRE sequences in their promoters as a transcription factor. This group of miRNAs is called “hypoxaMIRs” [60]. The best-characterized hypoxaMIR is mir-210, and it is considered to be the master of hypoxaMIRs [61]. Mir-210 has an HRE element 400 bp upstream from its core promoter. The HRE is highly conserved across species, suggesting that HIF-1α regulation has an important phylogenetic role. The evidence so far proves that mir-210 overexpression via HIF-1α favors cancer development [62]. In breast cancer, mir-210 expression is considered a biomarker of cancer progression, with a direct correlation to metastatic capability [63,64]. Another study showed that mir-210 is expressed in different cancer tissues (breast, lung, colon, pancreatic, head, and neck), and correlates with HIF-1α stabilization. Moreover, mir-210 can repress the expression of genes involved in the maintenance of normoxia, promoting hypoxic conditions in the tumor microenvironment [65]. Furthermore, mir-210 overexpression enhances proliferation and inhibits apoptosis in ovarian cancer [66], promoting vascular endothelial cell migration and angiogenesis in the brain [67]. Furthermore, it shifts the cancer cell metabolism, enhancing cell survival by driving tumor growth initiation [68]. These facts confer a pivotal relevance to mir-210, which acts as an oncomir, and highlights the importance of HIF-1α in cancer progression. 

In the same way that mir-210 has been described as the main hypoxaMIR in several types of cancer, there are other microRNAs reported as hypoxaMIRs. RNA sequencing technologies have revealed a considerable amount of information concerning miRNA profiles from a broad variety of cellular contexts, describing miRNAs regulated by HIF-1α [69]. Their understanding could contribute to developing new therapeutic strategies in cancer prognosis and treatment. Thus, we compiled a set of hypoxaMIRs regulated by the hypoxic response under some pathological conditions. These miRNAs were selected from 10 previous studies focused on hypoxaMIR identification, employing microarrays, RNA deep sequencing, and ChIP-seq. [58,61,70,71,72,73,74,75,76,77]. Subsequently, we created a list with all miRNAs obtained from these studies, and we investigated their role in the hallmarks of CRC (Table 1).

## 5. HypoxaMIRs Involved in CRC Hallmarks

### 5.1. Cell Cycle, Proliferation, and Apoptosis 

The control of cell proliferation has been linked to hypoxaMIR function in CRC samples. mir-21, mir-26a, mir-181a, and mir-103 promote tumor growth and enhance proliferation by downregulating PTEN expression; these microRNAs are also associated with clinicopathological features of CRC patients, being significantly overexpressed in patients with advanced clinical TNM stage. Interestingly, mir-103 regulates DICER and modifies the expression of microRNA profiles, increasing the proliferation rate in HCT116 cells. Therefore, restoring DICER and PTEN expression inhibits cell proliferation and induces cell cycle arrest in G0/G1 phase [78,79,80,81]. In addition, mir-103 also regulates proliferation by targeting LATS2 [82]. Meanwhile, mir-21 regulates tumor growth to alter protein traffic between the endoplasmic reticulum and Golgi apparatus via Sec23A [83]. Functional experiments demonstrated that locked nucleic acid treatment and overexpression of the long non-coding RNA LINC00312 directed against mir-21 reduce cell viability and decrease the invasive behavior in the LS174T cell line and in an in vivo nude mice model, respectively [84,85]. Another hypoxaMIR which is well-described in CRC development is mir-155, the expression of which increases the proliferation rate [86], thereby activating the PI3K-AKT pathway in HCT116 cells through PPP2CA (protein phosphatase 2A catalytic subunit alpha) and PTPRJ (protein tyrosine phosphatase receptor J) degradation, both negative regulators of this pathway [87,88]. In cancer cells, E2F2 binds to Rb1 protein in order to promote cell cycle progression, functioning as an oncogene. However, in colorectal cancer, it was reported for the first time that the E2F2 oncoprotein has a tumor suppressor role, but is negatively regulated by mir-155, having a direct impact on the proliferation of these cells [89,90]. Even though many earlier studies have described mir-181, mir-21, and mir-155 as direct targets of HIF-1α, we found reports that prove that these miRNAs also can be transcribed by NF-κB in CRC, which, in turn, can be positively regulated by HIF-1α during hypoxia and ischemia events, suggesting a key feedback loop that enhances cell proliferation and tumor growth [91,92,93]. 

Another hallmark of critical importance in the progression of cancer is the evasion of cell death. Apoptosis is necessary to maintain tissue homeostasis. In colon cancer cells, apoptosis participates in intestinal exchange, and its depletion promotes tumor transformation and progression [94]. Prior studies have indicated that multiple hypoxaMIRs reduce apoptosis via several mechanisms. mir-21 and mir-10b-5 repress RhoB [95] and PTEN expression [79,96] in HCT116, SW480, and HT29 cells, resulting in a significant decrease in apoptosis rates by annexin V detection, highlighting that mir-21 and mir-10b-5p increase cell proliferation through apoptosis inhibition. Similarly, mir-181b suppresses apoptosis through PDCD4 degradation [97] and the reduction in the NF-κB signaling pathway [98], leading to the modulation of a battery of key genes for the promotion of this cell death (Bax, caspase-3, and IκBα). Moreover, mir-27a-3p, which is highly expressed in both colon tissue and cell lines, abrogates apoptosis by targeting multiple genes involved in ERK/ MEK [99], Wnt/β-catenin [100], and apoptosome formation pathways (BTG1, RXRα and Apaf-1, respectively) [101]. In addition, other studies have shown that miR-155 exerts its oncogenic function by regulating the tumor suppressor gene FOXO3a, which transcribes caspase 3 and 7 [102]. Similarly, evidence supports that miR-107 contributes to the pathogenesis of human CRC by directly targeting Par4 [103], which increases Fas/FasL to the plasma membrane, and interacts with FADD activating caspase 8 and extrinsic apoptosis [104]. Therefore, reducing the expression of hypoxaMIRs involved in apoptosis is of potential interest in the search for chemopreventive and chemotherapeutic agents. Although extensive research is ongoing, the specific mechanism by which hypoxaMIR-mediated apoptosis is impaired remains unclear. All of the above provides a promising avenue for research.

### 5.2. Angiogenesis, Migration, and Invasion

HIF-1α mediates the transcription of VEGF to promote angiogenesis. The overexpression and activity of both are associated with tumor vascularization in CRC [37,105]. Nevertheless, at the time of writing, there is still not enough information regarding the regulation of angiogenesis via HIF-1α/hypoxaMIRs. The only related hypoxaMIR is mir-181, which has a critical role in the tube formation of endothelial cells, triggering VEGF secretion in HT29 and SW480 cancer cell lines [106]. Cancer progression depends on the metastatic and migratory potential of cancer cells, which relies on their expressing invasion and migration genes, and mir-21 appears to regulate this process [79,84,107]. Interestingly, different mir-21 isoforms have been found in CRC; there are 10 and 30 isoforms for mir-21-3p (passenger strand) and mir-21-5p, respectively. Due to this, mir-21 is also considered as an “isomir”, with each isoform showing different migration and invasion regulation abilities as a consequence of their different targets [108]. On the other hand, mir-155 enhances the invasion behavior of CRC cells increasing β-catenin mRNA and protein expression [109]. Moreover, mir-155 shows an interesting activity due to its sequence: it can increase RNA translation by binding AU-rich elements (AREs) in the 3′ UTR of its targets, effectively upregulating the expression of its targets. In this way, mir-155 regulates HuR (human antigen R) and RhoA expression positively to promote proliferation, migration, and invasion in CRC [110,111]. Another feature that allows cancer cells to migrate is filopodia formation. In CRC cells, the overexpression of mir-23a promotes the transition from indolent to invasive cancer through FBXW7 regulation. This gene induces the proteasomal degradation of c-MYC and c-Jun transcription factors. The downregulation of mir-23a via lentivirus transduction represses tumor formation and lung metastasis in a mouse model, and abolishes the presence of filopodia in CCSC cells [112]. Other studies have shown that mir-10b is upregulated in CRC patients with advanced clinicopathological features, and it was also found to be markedly elevated in lymph node and serum. Furthermore, the overall survival of CRC patients with high mir-10b levels is significantly shorter than that of patients with low mir-10b expression [113,114]. This hypoxaMIR acquires its function of increasing RhoC by targeting HoxD10; overexpression of RhoC enhances migration through the degradation and reconstruction of the extracellular matrix [115,116]. Hypoxia can lead to metastasis through specific mRNA translation control; PDCD4 functions as an eIF4A-dependent protein translation suppressor, and is recognized as a tumor suppressor protein. However, its function is inhibited by mir-181 overexpression. Interestingly, mir-181 is also transcribed by STAT3, suggesting that proinflammatory stimuli play a crucial role in metastasis signaling in correlation with HIF-1α activity. 

### 5.3. Metabolism and Inflammation

To maintain a high proliferation rate and growth, cancer cells have developed strategies to supply energetic requirements by changing their metabolism. In the presence of oxygen, normal cells process glucose in pyruvate via glycolysis. Thereafter, pyruvate is processed in the mitochondria via the tricarboxylic acid cycle, generating enough energy to maintain homeostasis. Nevertheless, cancer cells oxidize pyruvate in lactate, producing energy 100 times faster than normal cells. This phenomenon is known as the Warburg effect [117]. Under hypoxia conditions, RAS oncoprotein increases HIF-1α and HIF-2α transcription factors; in this manner, HIF-1α transcribes glucose transporters and multiple enzymes of the glycolysis pathway, enhancing the Warburg effect. Another mechanism that involves lactate production is mediated by mir-26a. This hypoxaMIR regulates pyruvate dehydrogenase protein X complex (PDHX), avoiding pyruvate entrance in the tricarboxylic acid cycle; therefore, whole pyruvate production is metabolized in lactate [118]. Changes in metabolism are also associated with chemoresistance. In CRC hypoxic microenvironments, HIF-1α overexpresses mir-21 and mir-30d, altering amino acid metabolism. As a consequence, CRC cells treated with 5-FU obtained chemoresistance; as such, the inhibition of these hypoxaMIRs brings them back to drug sensitivity [119]. Moreover, metabolism is a pivotal regulator of gene expression due to the metabolites produced by these metabolic pathways being used as cofactors, regulating epigenetic processes, such as DNA and histone methylation. Folate is the key methyl group donor, and it is related to mir-21 overexpression in serum, serving as a CRC biomarker [120]. Moreover, aberrant metabolism can influence changes in the inflammatory process. For example, high levels of glucose uptake trigger pro-inflammatory activation of macrophages through the production of reactive oxygen species. 

Chronic inflammation is one of the principal etiopathological features that promote CRC development; ulcerative colitis is considered a premalignant condition. Inflammatory stimuli produce reactive oxygen species (ROS) that damage DNA, leading to DNA mutations. In addition, chronic inflammation allows cytokine, chemokine, and growth factor production, which cause oxidative damage and epigenetic silencing of tumor suppressor genes [9]. Moreover, miRNAs have been implicated in the innate immune response and adaptive immunity, since they control T- and B-cell maturation [121]. The AOM/DSS CRC inflammation model represents the best model symptoms associated with this disease. mir-26b was found to be overexpressed in the serum of AOM/DSS-treated mice, as compared to controls. The results showed that, when mir-26b was coimmunoprecipitated with AGO2, there was significant enrichment of DIP1, MDM2, BRCA1, PTEN, and CREBBP mRNAs. Such mRNAs are involved in apoptosis induction, avoiding proliferation, and stopping pro-inflammation processes, suggesting that mir-26b may coordinate cross-talk of different pathways [122]. Associated with the inflammatory process, it has been reported that overexpression of mir-21, IL6, and IL-8 in peripheral blood-derived plasma is negatively associated with relapse-free periods and overall survival in patients with metastatic colorectal cancer [123]. Another critical event in CRC development is the evasion of the immune response. This process depends on antigen processing and the presentation by of histocompatibility complex 1 (MHC1); downward activation of MHC1 is also detected in 74% of cases of CRC. For MHC1 to exert its function, it must be exposed to the cell surface, but in order for this molecular event to happen, it is necessary for it to be assembled with calnexin and calreticulin protein chaperones. However, it has been reported that mir-27a directly regulates these mRNAs, thereby preventing MHC1 exposure [124]. 

MicroRNAs are also expressed in many immune cells; the imbalance in the expression of microRNA profiles changes their behavior and function. One example is mir-24; its overexpression in natural killer cells downregulates Paxilin levels, altering their killing effect to CRC cells [125], despite documented evidence that macrophages M2 can promote cell migration and invasion in CRC through microRNA delivery via exosomes [126]. These facts show us how hypoxaMIRs modulate the immune response in CRC. Figure 3 summarizes the role of hypoxaMIRs in colorectal cancer.

With all of the evidence described above, we can demonstrate the potential value of hypoxaMIRs as master regulators of each hallmark in CRC, as well as evidence of the major tumor suppressor genes that are affected by hypoxia. Furthermore, by means of data mining of CRC patients and analysis of interaction networks, which are described in detail in the following subtopic, we corroborated that HIF-1α has a high clinical value in colon tumor progression.

## 6. HypoxaMIR Expression in TCGA Database and Its Correlation with the Hallmarks of Cancer

With the attempt to understand the biological impact of hypoxaMIRs on the molecular mechanism involved in CRC, we analyzed the expression levels of hypoxaMIRs in the TCGA database (Table 1). miRNA-seq reads data were obtained using the Bioconductor TCGA biolinks package [127], and differential expression analysis between normal and tumoral tissues was assessed using the DESeq2 package [128], considering only miRNAs with an FDR < 0.05 as statistically significant (Figure 4). These data show that hypoxaMIRs are highly expressed in colorectal cancer tissue (red), compared with normal samples (light blue).

We randomly selected six hypoxaMIRs (mir-19b, mir-21, mir-24, mir-26b, mir-101, and mir-106b), and their targets were predicted using the multiMiR package [129]. Next, we took targets that were present in the Catalogue of Somatic Mutations (COSMIC) database [130]. We found that all of these miRNAs mainly have target genes involved in the suppression of growth, escaping of programmed cell death, and invasion and metastasis (Figure 5), thereby supporting the relevance of hypoxaMIRs expression in colorectal cancer progression and development. 

## 7. HypoxaMIRs and Its Relevance as Biomarkers in CRC Clinical Outcomes

The necessity of novel markers or therapeutic targets in cancer has brought hypoxaMIRs forward as excellent candidates. Since 2012, before being considered, several HIF-regulated miRNAs have been used as biomarkers; for example, let-7i was found to be differentially expressed in several CRC monolayer cell lines [131]. Moreover, mir-155 was measured by stem-loop qPCR in 109 pair-matched human CRC samples and the corresponding normal mucosa, showing the same findings as let-7i [132]. One of the main molecular events in CRC development is K-RAS overexpression. As such, Ota and collaborators developed a K-RAS variable expression model in 3D culture, and their results indicated that mir-181a and mir-210 were significantly overexpressed in a DLD-1 3D culture versus a DLD-1 monolayer culture, as well as in human CRC tissues [133]. These results suggested that hypoxaMIR expression variates even in the same CRC cell lines and, most likely, in a human sample. Over time, various hypoxaMIRs have been selected as reliable markers; they have been measured in different histological cancer stages, and have also been associated with clinical outcomes. Such a finding allowed for new miRNA signatures (mir-193a, mir-23a, mir-338-5p, and mir-10b) to be involved in cancer progression [134,135]. Furthermore, the prognostic value of hypoxaMIRs in chemotherapy response has been successfully tested. For example, in a cohort of 78 CRC patients, low mir-429 expression was associated with a positive response to 5-FU treatment; in contrast, high mir-429 expression was correlated with poor prognosis, metastatic phenotype, and non-5-FU response [136]. Moreover, many studies have matched hypoxaMIR availability in different fluids with clinicopathological features. For example, they have been isolated from circulating plasma, and associated with treatment response [137] and poor prognosis [138], as well as having been isolated from serum in association with CRC development [139]. All of these lines of evidence support the value of hypoxaMIRs as a prognostic marker during colorectal cancer progression in each of the cancer hallmarks (Figure 6). 

## 8. Concluding Remarks

Here, we have analyzed a group of microRNAs, known as hypoxaMIRs, that are directly transcribed by the transcription factor HIF-1α. Together, these hypoxaMIRs which are overexpressed in CRC participate in different hallmarks of cancer. Furthermore, by analyzing their expression and participation in the establishment of the tumor phenotype, we show that a select group of them (miR-19b-3p, -21-5p, -24-3p, -26b-5p, -101-3p, and -106b-5p) mainly regulate invasion and metastasis, key events in advanced clinical stage tumors.

## Figures and Tables

**Figure 1 cells-11-01895-f001:**
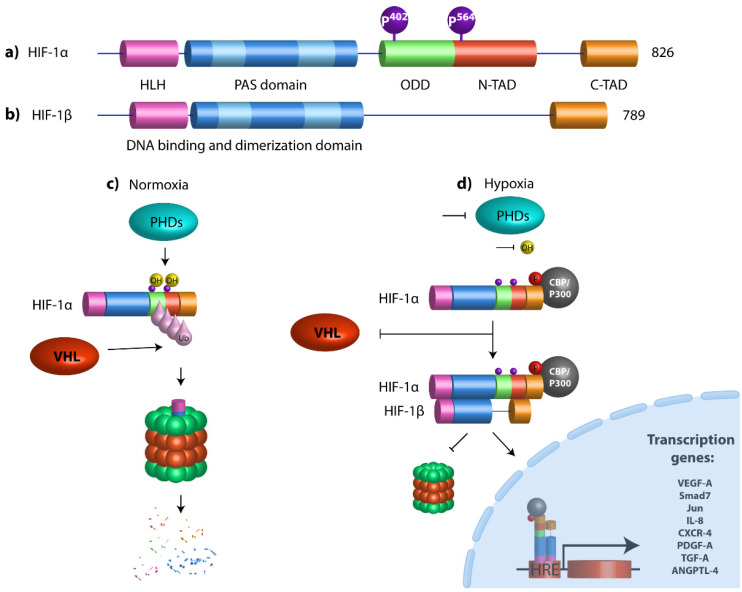
Structural domains of HIF-1α and HIF-1β and schematic diagram of HIF pathway. (**a**) HIF-1α contains a basic helix–loop–helix (HLH) system and two dimerization domains (PAS-A and PAS-B) that mediate DNA binding and dimerization, respectively. An oxygen-dependent degradation domain (ODD) is required for oxygen-dependent hydroxylation and degradation under normoxia conditions. The transactivation domains (N-and C-Terminal TADs) are both responsible for the transcriptional activity. N-TAD domain is located within the ODD domain and C-TAD domain at the C-terminal region of the protein. Prolyl hydroxylases (PHDs) hydroxylate HIF-1α proline residues 402 and 564. (**b**) HIF-1β is the aryl hydrocarbon receptor nuclear translocator (ARNT), which has a bHLH domain, PAS-A and PAS-B domains, and only one C-terminal transactivation domain. (**c**) Under normoxic conditions, HIF-1α undergoes hydroxylation by prolyl-4-hydroxylases (PHDs) at proline residues 402 and 564, which promotes the destabilization of the HIF-1α protein, which allows the von Hippel-Lindau (VHL) protein to bind it, resulting in polyubiquitination, leading to proteasome-mediated degradation. (**d**) Under hypoxic conditions, PHDs are inactivated, promoting HIF-1α stabilization. HIF-1α dimerizes with its partner HIF-1β (ARNT), they are translocated to the nucleus, and, in combination with the transcriptional coactivator CBP/P300, they bind to genomic DNA at hypoxia response elements (HREs) to activate the transcription of target genes.

**Figure 2 cells-11-01895-f002:**
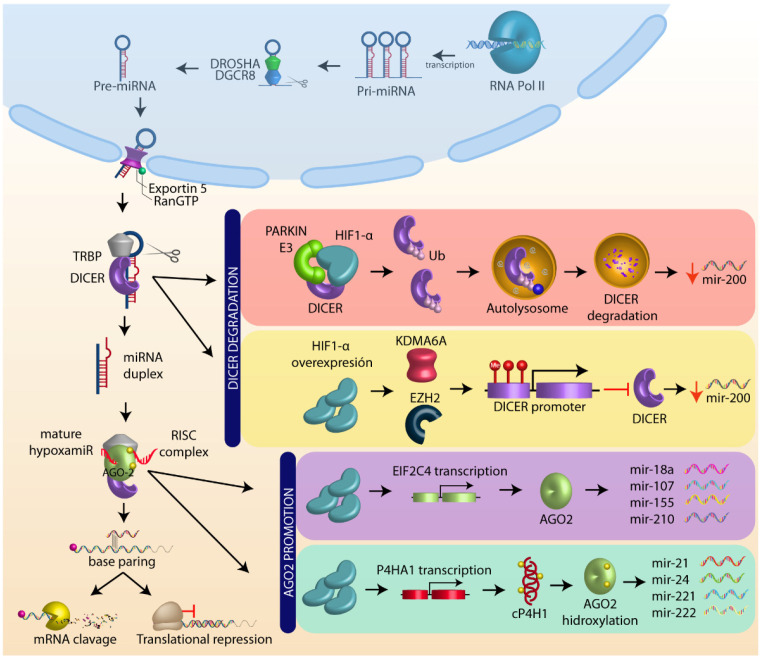
MicroRNA biogenesis and HIF-1α role. MicroRNA (miRNA) are transcribed as primary miRNAs (pri-miRNAs) by RNA polymerase II (Pol II) in the nucleus. The pri-miRNAs are cleaved by the endonuclease DROSHA/DGCR8 (DiGeorge syndrome critical region 8) to a shorter pre-miRNA hairpin structure (60–70 nucleotides). The pre-miRNAs are exported from the nucleus to the cytoplasm by exportin-5-Ran-GTP (XPO5), and are processed by DICER1/TRBP, a ribonuclease III (RIII) enzyme that produces miRNA duplex (~21-nucleotide). The final step of miRNA maturation is the selective functional strand of small RNA duplex into RNA-induced silencing complex (RISC), which includes DICER, TRPB, and Argonaut (AGO). One strand of the mature miRNA (the guide strand) is loaded with Argonaut (AGO2) to form a miRNA-induced silencing complex (miRISC) that targets mRNAs by a complementary sequence leading to mRNA degradation and translational repression. HIF-1α negatively regulates DICER through autolysosome degradation and promoter methylation, whereas HIF1-α positively regulates AGO by target transcription and promotes AGO hydroxilation (see text).

**Figure 3 cells-11-01895-f003:**
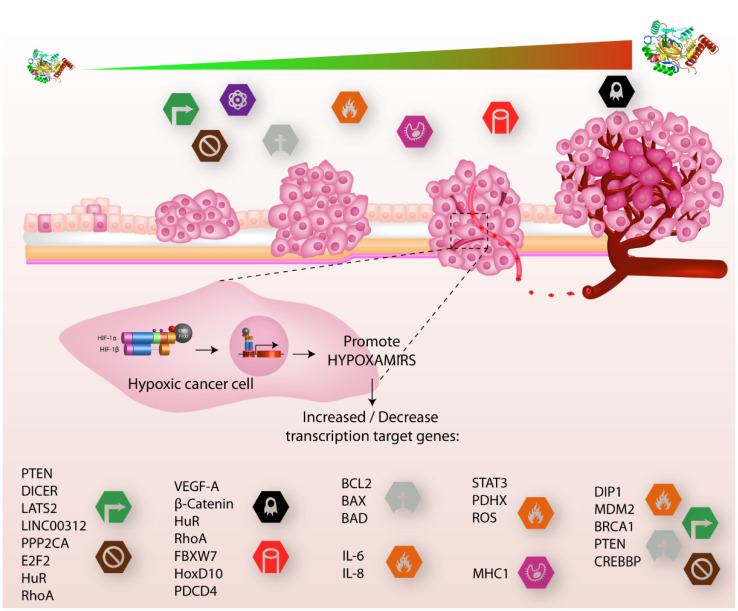
HIF-α involved in cancer progression. In the early stages of tumor growth, cells acquire distinctive hallmarks that promote cancer progression, such as sustaining proliferative signals, evasion growth suppression, reprogramming energy metabolism, and resistance to cell death, while in the final stages of tumor growth, hallmarks which are added include inflammation, avoiding immune destruction, and, finally, metastasis and angiogenesis. These features are regulated by different genes (lower part) that are activated or inhibited by different hypoxaMIRs, which are regulated positively and negatively by the transcription factor HIF1-α, which, in turn, increases the concentration of proteins as tumor progression increases.

**Figure 4 cells-11-01895-f004:**
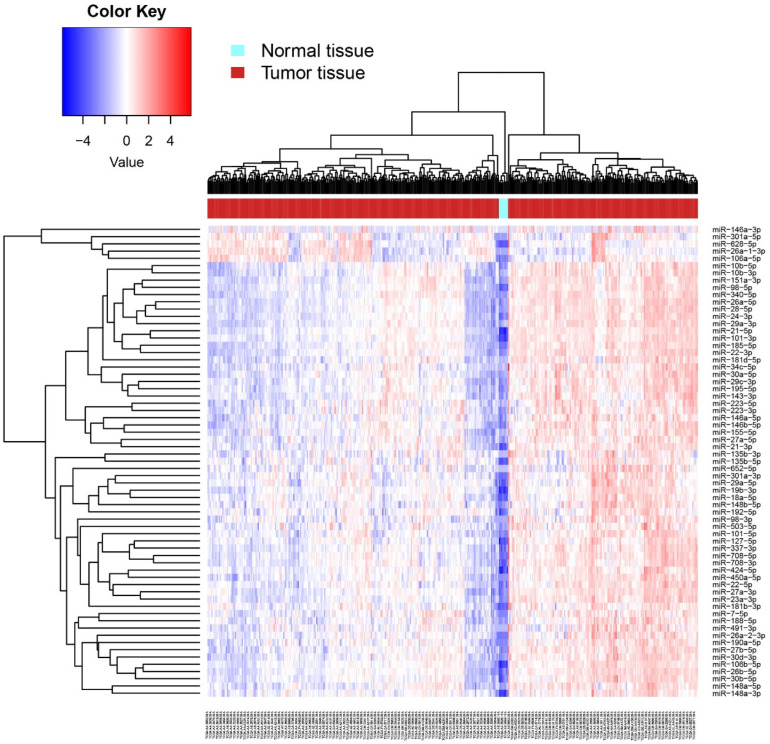
Heat map of differential expression of hypoxaMIRs in human normal and tumor tissues. Differential expression of hypoxaMIRs was identified in normal tissue and tumor tissue by a miRNA body map along with the hierarchical cluster analysis. Expression of hypoxaMIRs is represented as blue (downregulated), red (upregulated), and white colors (no significant change or absence of data).

**Figure 5 cells-11-01895-f005:**
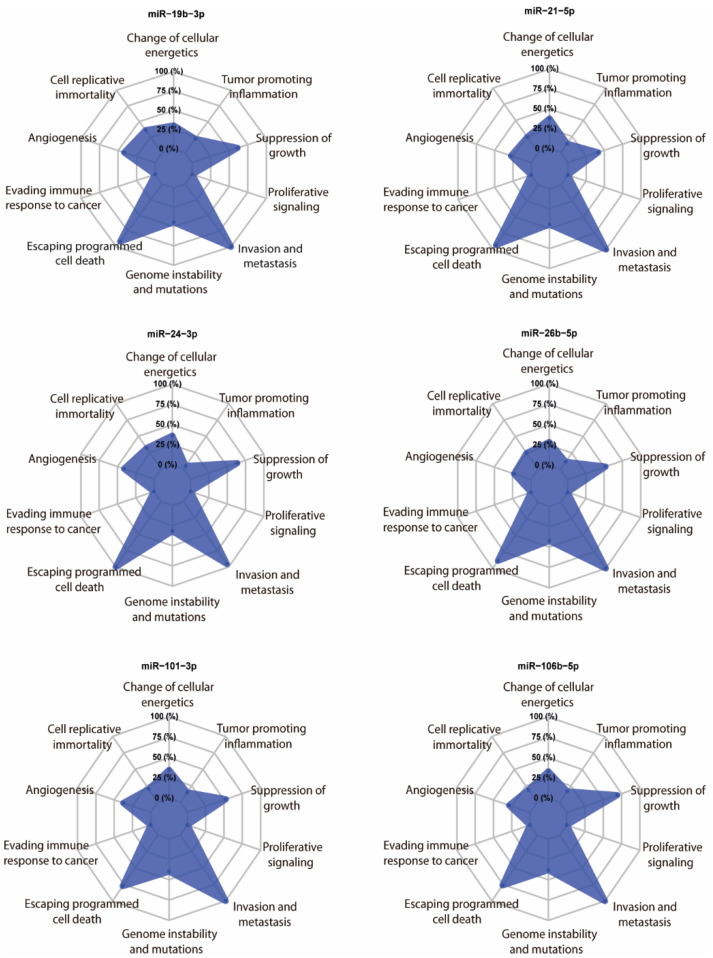
In silico analysis. Radar chart of the relevant hallmarks of cancer simultaneously modulated by the selected hypoxaMIRs after analysis. The levels show the proportion of the number of putative gene targets for each class.

**Figure 6 cells-11-01895-f006:**
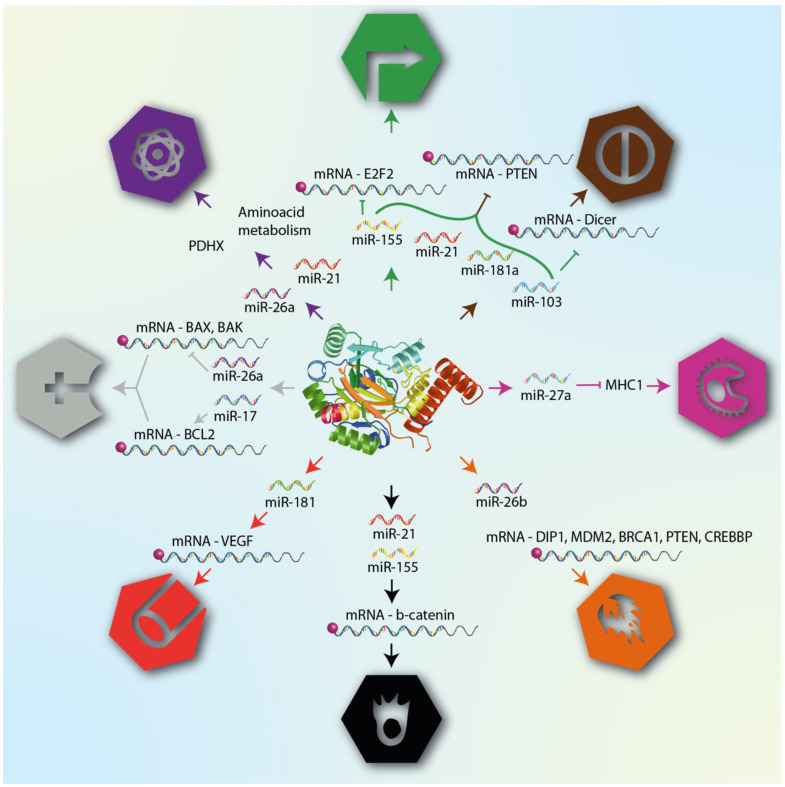
HypoxaMIRs associated with their corresponding cancer hallmark. The figure lists some hypoxaMIRs (previously described in this review) that regulate genes involved in the different hallmarks of cancer, as defined by Hanahan and Weinberg. These hallmarks are cellular mechanisms acquired during tumor transformation, and include proliferation, apoptosis, angiogenesis, invasion, and metastasis.

**Table 1 cells-11-01895-t001:** microRNAs regulated by HIF-1α in hypoxic response. This table represents those microRNAs that were regulated under hypoxic conditions (upregulated or downregulated). We selected those microRNAs that appeared in previously mentioned studies. The asterisk represents the miRNA presence in each of the articles reviewed; thus, a miRNA with six asterisks mean that this molecule was present in six of the articles reviewed, and so on. On the other hand, a miRNA without asterisk means that was present in just one report.

Hypoxamirs						
Upregulated by Hypoxia					Downregulated by Hypoxia	
mir-210 *****	mir-130 *	mir-133b	mir-301b	mir-539	mir-15b **	mir-331
mir-21 ****	mir-148b *	mir-135b	mir-30a	mir-563	mir-16 **	mir-422b
mir-26a ****	mir-151a-3p *	mir-143	mir-30d	mir-572	mir-200b **	mir-449a
Let-7e ***	mir-181a *	mir-146a	mir-322	mir-596	mir-199a *	mir-449b
mir-103 ***	mir-181b *	mir-146b	mir-323a-3p	mir-628	mir-20a *	mir-484
mir-107 ***	mir-185 *	mir-148a	mir-324-3p	mir-637	mir-378 *	mir-551b
mir-199a-5p ***	mir-192 *	mir-149	mir-337-3p	mir-652	mir-112	mir-565
mir-23b ***	mir-19b *	mir-15	mir-339-5p	mir-664	mir-122a	mir-584
mir-26b ***	mir-213 *	mir-181d	mir-340-3p	mir-696	mir-1255b	mir-589
mir-424 ***	mir-22 *	mir-188	mir-340-5p	mir-699	mir-135a	mir-622
Let-7b **	mir-320 *	mir-18a	mir-342	mir-708	mir-141	mir-877
mir-125b **	mir-373 *	mir-190a	mir-345-3p	mir-769	mir-144	mir-92
mir-155 **	mir-429 *	mir-191	mir-345-5p	mir-99	mir-150	mir-96
mir-204 **	mir-433 *	mir-194	mir-34c	Let-7a	mir-17	
mir-23a **	mir-503 *	mir-195 **	mir-370		mir-181a	
mir-24 **	mir-7 *	mir-205	mir-372		mir-18b	
mir-27a **	mir-101	mir-214	mir-450a		mir-193b	
mir-30b **	mir-106a	mir-221	mir-451		mir-197	
mir-93 **	mir-1180	mir-223	mir-466d		mir-19a	
mir-98 **	mir-1185	mir-27b	mir-491		mir-200c	
Let-7c *	mir-127	mir-28	mir-497		mir-202-3p	
Let-7i *	mir-128a	mir-298	mir-498		mir-20b	
mir-1 *	mir-129-2	mir-29a	mir-500a		mir-224	
mir-106b *	mir-132	mir-29c	mir-532-3p		mir-29b	
mir-10b *	mir-133a	mir-301a	mir-532-5p		mir-30e	

## Data Availability

Not applicable.

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
