# Peer review of "HypoxaMIRs: Key Regulators of Hallmarks of Colorectal Cancer"

_cells, 2022, doi:10.3390/cells11121895_

Round 1

Reviewer 1 Report

Update reference 1 of 2018 (Global Cancer Statistics 2018: GLOBOCAN) with that of 2020.

Lines 64-65: incomplete sentence "molecular marker of."

Lines 85: correct punctuation in the sentence "HIF-1α recognizes and binds to hypoxia response elements (HRE the consensus 84 sequence is G / ACGTG) (Figure 1), [26]; present in the promoter region "

Line 145: insert full stop after [41]

Line 231: "induces" instead of "induce"

Line 256: remove the semicolon in [72]; [89]

Line 346: write the sentence better: "overexpression of mir-21, IL6, and IL8 had been used as circulating inflammation signature that difference between overall survival and relapse-free survival in patients with metastatic CRC "

Line 355: correct "unbalanace"

Line 373: correct "mechanisim”

Line 404: correct "ann”

Line 434: incomplete sentence

Author Response

Reviewer 1

Update reference 1 of 2018 (Global Cancer Statistics 2018: GLOBOCAN) with that of 2020.

Dear reviewer, thank you very much for your thoughtful observation, the reference was updated as you suggested.

Lines 64-65: incomplete sentence "molecular marker of."

Dear reviewer, the sentence was corrected and replaced by following statement “Therefore, even though the relevant role of HIF-α activation in cancer development is not a bona fide molecular marker”

Lines 85: correct punctuation in the sentence "HIF-1α recognizes and binds to hypoxia response elements (HRE the consensus 84 sequence is G / ACGTG) (Figure 1), [26]; present in the promoter region "

The sentence was replaced by the following one: HIF-1α recognizes and binds to the consensus sequence G/ACGTG better know as HRE (hypoxia response elements) [26]

Line 145: insert full stop after [41]

Full stop was corrected as you suggested

Line 231: "induces" instead of "induce"

It was corrected

Line 256: remove the semicolon in [72]; [89]

It was corrected

Line 346: write the sentence better: "overexpression of mir-21, IL6, and IL8 had been used as circulating inflammation signature that difference between overall survival and relapse-free survival in patients with metastatic CRC "

The sentence was replaced by the following: I”t has been reported that overexpression of mir-21, IL6 and IL-8 in peripheral blood-derived plasma is negatively associated with relapse free and overall survival in patients with metastatic colorectal cancer

Line 355: correct "unbalanace"

The sentence was corrected.

Line 373: correct "mechanisim”

The typo was corrected.

Line 404: correct "ann”

The mistake was corrected.

Line 434: incomplete sentence

The sentence was corrected.

Reviewer 2 Report

Since the review article describes a potentially interesting hypoxia-related miRNAs in colorectal cancer, I would propose to accept the revised version of the manuscript, which addresses the major concerns.

Major concerns:

-       Since the structure of the review article is complicated, it is better to describe each headline clearly.

-       It is better to describe the relationship between Hypoxia-related miRNAs and drug resistance.

-       -It is recommended to briefly explain drug resistance in headline 2 "Hypoxia-inducible factors".

Minor concerns:

-       The following abbreviations should be spelled out in the text. CCR (lines 45 and 56), HLH (line 73), and PAS (line 73).

-       Check the text on line 65. Especially about the positions of “.” and “,”.

-       The authors need to check the entire text.

Author Response

Reviewer 2

Major concerns:

-       Since the structure of the review article is complicated, it is better to describe each headline clearly.

Dear reviewer, we have changed  some of the headlines, hence we consider they are clearer.

-       It is better to describe the relationship between Hypoxia-related miRNAs and drug resistance.

Dear Reviewer, the aim of this manuscript was to show the roloe of HIF-1a as a master regulator of transcriptional activity of some miRNAs collectively called “Hypoxamirs” and how they can regulate many hallmarks in CRC. Although, how these microRNAs influence drug resistance it is an interesting topic; we consider that it deserves the attention to another singe manuscript. However, we have incorporated to the discussion of the headline 2 some examples of these resistance mechanisms.

-       -It is recommended to briefly explain drug resistance in headline 2 "Hypoxia-inducible factors".

       Dear Reviewer, we have added the following examples to address the relevance of drug resistance  mechanisms activated by HIF:

Moreover regulating the expression of many genes involved in tumor maintenance, it has been described that HIF-1a is able to promote drug resistance through metabolic reprogramming by increasing the transactivation of PDK1, LDHA and BNIP3-L to switch from oxidative to glycolytic metabolism, an event known as the Warburg effect [17]. It has been reported that this metabolic rewire prevents mitochondrial production of reactive oxygen species reducing the DNA damage and attenuating the antitumor effect of TKIs (Tyrosin-Kinase Inhbitors) in several types of cancer [30][31][32][33]. Besides, in CRC cells treated with 5-FU (5-fluoruracil), the damage of mitochondria promotes the loss of their main source of energy and consequently cell death is induced. In this scenario HIF-1a through the activation of the Wnt/b Catenin and PI3K pathways, stimulates the transcription of glycolytic genes such as GLUT1, HK2, PKM2 and LDHA, restoring the lost energy and promoting drug resistance to 5-FU [34].

CRC cells develop drug resistance by different mechanisms; patients bearing the G12V mutation in the KRAS gene it has been shown that high levels of HIF-1a are associated with resistance to Cetuximab. When a HIF inhibitor is used (PX-478), the cells were drug-sensitized, abating proliferation [35]. In addition, CRC-derived cell lines treated with DNMTs inhibitors, indirectly diminished HIF-1a activity, leading to loss of resistance to oxalaplatin treatment [36]. Therefore, due to the large number of mechanisms involved in cancer establishment regulated by HIF-1a, it is important to describe their relevance in colorectal cancer.

Minor concerns:

-       The following abbreviations should be spelled out in the text. CCR (lines 45 and 56), HLH (line 73), and PAS (line 73).

Dear reviewer, the sentence was corrected as you suggested

-       Check the text on line 65. Especially about the positions of “.” and “,”.

Dear reviewer, the sentence was corrected as you suggested

-       The authors need to check the entire text.

Dear reviewer, we have carefully checked the manuscript as you have suggested, and we consider that the final version is ready and corrected.

Round 2

Reviewer 2 Report

The manuscript has been revised well. I recommend that it be accepted for publication.